# Maximal response to a mechanical leader at critical group size in ant collectives

Atanu Chatterjee [1] ✉, Tom Tzook[1], Nir S. Gov [2] & Ofer Feinerman[1] ✉

It is widely recognized that biological collectives operate near criticality to amplify their capability of collective response. The peak in susceptibility near criticality renders these groups highly responsive to external stimuli. While this phenomenon has been recognized and supported by evidence from theory, a direct experimental demonstration has been elusive. To bridge this gap, here we record the response of a group of *Paratrechina longicornis* ants to external stimuli as they join efforts to carry food to their nest. Using a robotic system that mimics a transient leader, we apply tactile ant-scale forces and measure the group's response at sub, near, and supercritical regimes. Supported by theory and simulations, we provide direct experimental evidence to demonstrate that at critical group size, the collective response of the ants to an external force is maximally amplified.

Collective behavior in biological systems represents one of nature's most fascinating and ubiquitous phenomena. From the flocking of birds and schooling of fish to the coordinated firing patterns in the brain, these ensembles are a testament to the complexity of life[1–8]. Exploring these systems offers more than just an understanding of biological complexity; it provides a distinctive perspective on the intricate interactions that dictate the behavior of wholes that are made up of numerous interacting components. A key characteristic of biological groups is that they display emergent collective states that are often highly coordinated and display long-range order[1–3]. It has been shown that some groups can maintain multiple distinct collective states and can switch between these states depending on group size, density, boundary conditions, or random fluctuations[7,9–13]. Moreover, emerging evidence suggests that many of these systems operate near the transition between different collective-level states[14–21].

Over the past decades, statistical physics has offered tools for studying self-organization and the emergence of order in biological collectives[14,15,22]. Additionally, physical phases provide intuition regarding the coexistence of different collective states in a single biological system. Even more interesting, statistical physics offers possible reasoning for the fact that biological ensembles are poised at a transition between collective states. Indeed, it is known that, in such transition regions, physical systems display critical phenomena such as long-range correlations and increased susceptibility to external perturbation[23,24]. These properties are interesting from a biological

perspective since they allow the group to react quickly and cohesively to changing environmental conditions[20,22,25,26]. Consequently, studies ranging from regulatory interactions among proteins in cells to neurons interacting in brains to collective behavior in social insects, fish schools, bird flocks, humans, and mammals have hypothesized that complex biological collectives might operate near criticality to enhance sensitivity to external stimuli and maximize collective computational capabilities[14,19,21,22,26–31].

Despite these theoretical predictions, experimentally testing the hypothesis that complex biological collectives may operate near criticality has been very challenging. Bridging the gap between theoretical predictions and empirical evidence encounters a substantial hurdle due to the lack of direct measurements of how a biological group responds to changes in a specific control parameter close to the transition regime. This challenge is only compounded by the critical point being precisely defined only in the thermodynamic limit[23,24]. Additionally, there is evidence that many biological collectives may override control parameters and continuously self-tune to remain near a critical-like state, making it difficult to probe the transition between the different phases[7,14,15,20,30,32,33]. Finally, identifying what precisely stimulates a biological collective poses an additional layer of complexity. The potential cues can widely range from visual to chemical, tactile to auditory, or even a combination of these factors[34–37]. As a result, although indirect evidence suggests increased susceptibility in biological ensembles near the transition between collective states, direct

[1]Department of Physics of Complex Systems, Weizmann Institute of Science, Rehovot, Israel. [2]Department of Chemical and Biological Physics, Weizmann Institute of Science, Rehovot, Israel. ✉e-mail: achatterjee.physics@gmail.com; ofer.feinerman@weizmann.ac.il

empirical evidence remains very limited and sparse. Therefore, while criticality offers a compelling framework for understanding the relationship between self-organization and biological functionality, it remains a topic of ongoing investigation that requires further empirical validation across diverse biological contexts[38,39].

In this work, we provide the first direct demonstration of increased susceptibility in the transition between two collective states. We do this in the context of cooperative transport by longhorn crazy ants (*Paratrechina longicornis*), where we show that when the groups' collective motion is at the transition between distinct collective states, it is most sensitive to single-ant scale perturbations. More specifically, during cooperative transport, these ants gather to collectively haul a load that is too heavy for any of them to move on their own[40,41]. It has been previously shown that the hauling group can assume one of two collective states – one in which the load moves ballistically and another where the collective motion resembles a random walk[10,31]. Further, it has been demonstrated that, in natural conditions, the group resides near the transition between these two order-disorder collective states. The biological relevance of this is crucial, as its criticality and enhanced sensitivity allow carrying groups to be maximally responsive to information brought in by spatially informed leader ants and successfully navigate toward their nest[10]. Finally, cooperative transport in this species can be mapped to excellent quantitative and qualitative agreement to an Ising spin model. This mapping reveals that the transition between collective motion states in the finite-sized ant system corresponds to the critical transition between magnetized and non-magnetized states in the Ising model, which represent the ordered and disordered phases of the ants, respectively, in the thermodynamic limit ($N \to \infty$)[10,24,31].

The ant system is unique as it circumvents many of the obstacles mentioned above and provides a *rare* opportunity to measure increased susceptibility directly when the system transitions between the two collective states. First, crazy ants that engage in cooperative transport do not display self-tuned criticality. Instead, a simple control parameter—group size—can drive the group from a disordered collective state through the transition regime to an ordered state[10,31]. Moreover, the small perturbation to which the ants respond is clear. During cooperative transport, ants in the group are coupled to each other through forces that are transmitted via the jointly carried load. Therefore, if we wish to measure susceptibility to small perturbations as a function of group size, we apply a small force to the carried load[10,11,31]. Finally, these measurements can all be performed in the field, where the animal group displays its natural behavior.

In this work, we leverage these advantages by employing a novel robotic system that mechanically applies millinewton scale forces to a load as it is hauled by a group of ants. The robot mimics the force applied by a transient leader ant that injects directional information into the group. Importantly, the control parameter, group size, is varied simply by using loads of different sizes, which allows us to observe and quantify group susceptibility to a small external force across various collective states.

## Results

### Force application robot

We attached the cargo to a stiff rod hinged to a pivot, as shown in Fig. 1. The pivot confines the cargo to move along a circular path determined by the length of the rod, $l_{rod}$. Varying cargo radius provides control over the size of the carrying group. We recorded cooperative transport for four group sizes: small ($N = 2-4$ ants), intermediate ($N = 15-20$ ants), large ($N = 30-40$ ants), and very large ($N = 50-60$ ants), where $N$ represents the number of ants attached to the cargo. We observe a transition between two modes of collective motion as a function of the group size (Fig. 2). The cargo frequently switched directions for the smallest groups and displayed largely erratic movements, primarily due to tug-of-war among the few ants pulling the cargo. In contrast,

larger groups displayed a smooth and ordered collective motion with persistent pendulum-like oscillations. These two phases become evident when considering the order parameter, which, in this system, is the load's angular velocity. Indeed, the order parameter changes its distribution from a single peak distribution centered around zero in the disordered phase to a bimodal distribution peaked at the two values of the spontaneous velocity (analogous to magnetization in the Ising model) in the ordered phase (top panels of Fig. 2). Importantly, the transition between disordered and ordered motion is observed to be around the intermediate group size when the group size is on the order of ten ants. These results are in agreement with previous work on cooperative transport of a hinged load[11]. This transition is also exhibited by plotting the angular velocity as a function of the angular position for different group sizes (Fig. S2 in the Supplementary Text).

In contrast to previous work where the hinged load system was passive and allowed for free rotations[11], in this study, we employed an active "robot"[42] capable of reacting to the ant's motion and applying external tangential forces to the load (see Fig. 1). Using feedback from the position sensor at the pivot axis, the robot continuously monitors the ants' angular position and rotates its blade according to this angle as well as pre-programmed commands (Fig. 1B). Force application is achieved via a soft cantilever that is mechanically connected to the load and and rotates with it (Fig. 1C). Along with the blade screws shown in Fig. 1C, this cantilever functions as a clutch for the force application system. To allow the ants to move freely, the robot continuously adjusts the blade's position such that the cantilever arm remains situated between the blade screws. This effectively disconnects the robot from the cargo. Upon receiving instructions to exert an external force, the robot moves its cantilever slightly off-center so that one of the blade screws gently pushes on the soft cantilever. The left screw allows for application of a clockwise force, and the right screw for a counterclockwise force. The angular deformation experienced by the soft cantilever is then transmitted on the cargo as a tangential force, $F_{ext}$, which persists for a fixed duration, $\Delta t$. This novel setup allows precise control over the location, duration, and strength of the external force. Further details on the algorithm used to apply forces using the robot can be found in the Supplementary Text.

For this work, the robot was used to apply a single force pulse whose magnitude was on the scale of a single ant force. The external force was applied when the cargo was closest to the nest, at the midpoint of an oscillation, $\theta = 0°$. The force was exerted in the direction that opposes the ants' collective motion. Thus, the robot mimics an informed ant, assumes the role of a mechanical leader, and provides the group with useful directional information. This force works to reverse the direction of the collective motion and corrects it to be directed toward the nest. Field recordings of the robot perturbing cooperative cargo transport are shown in Movies S1 (resist) and S2 (switch).

The robot was programmed to exert a force pulse of specific magnitude and duration. However, since the forces involved are minute and the closed-loop system includes the unpredictable ants, we did not base our analysis on these pre-programmed parameters; instead, we measured each pulse using image analysis. We present a sample experimental run to demonstrate the force application and measurement process. Figure 3A shows the tracked angular positions of both the ant-held cargo (group size of $N = 50-60$ ants) and the robot's force application blade. The difference between these two angles can be calibrated (see, Supplementary Text Fig. S1) to produce a time series of the external forces that the robot exerts on the ants (Fig. 3B). Last, we approximated this force timeline by a series of rectangular pulses. A group may either resist the force or switch its alignment in response to the external force. If the group resists, the total duration of the force application is set as $\Delta t = (t_{end} - t_{begin})$ (where $t_{begin}$ marks the time at which force application commenced and $t_{end}$ the time in which it ended).

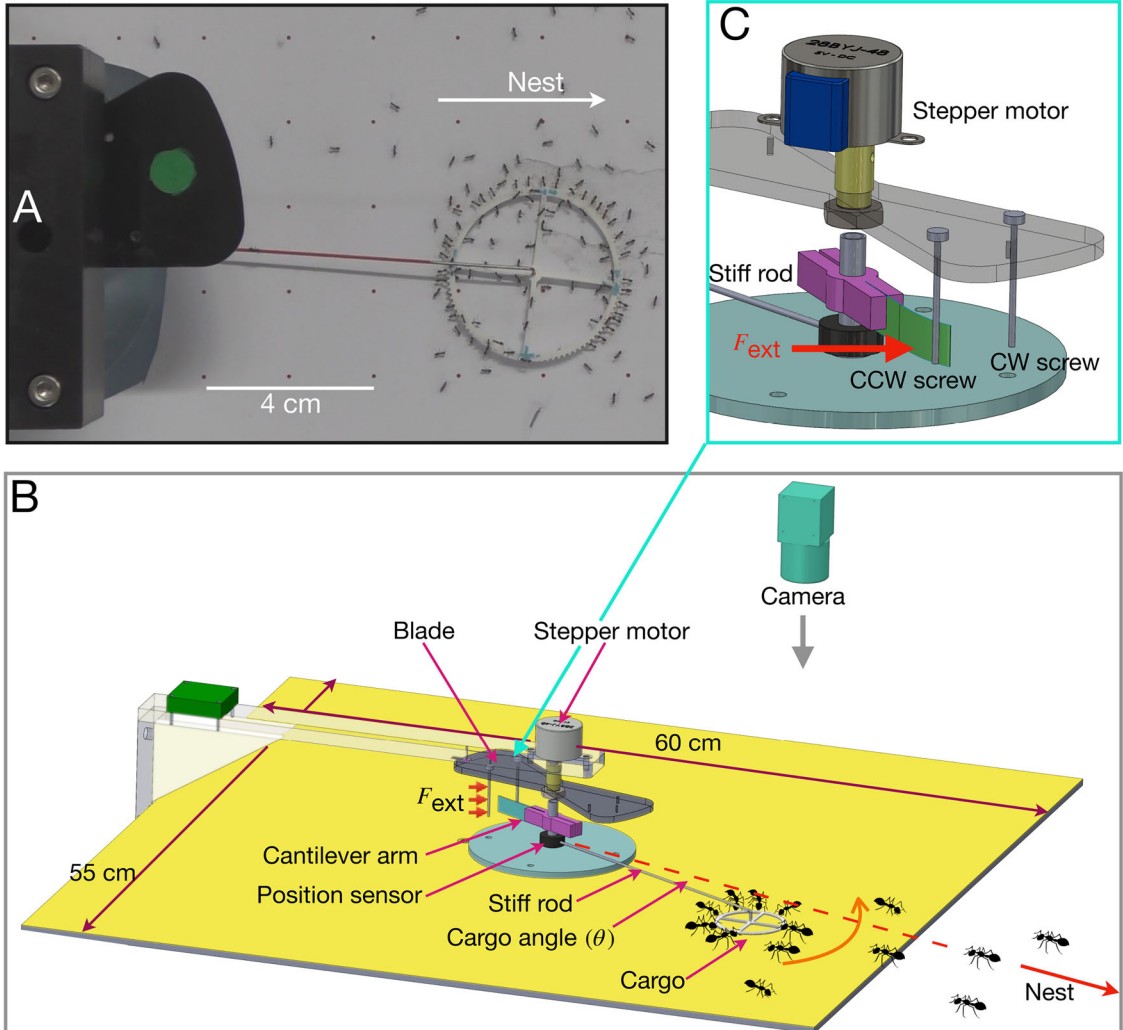

**Fig. 1 | Cooperative transport with a mechanical leader. A** Movie frame recording of the cooperative transport in action. The silicone ring cargo is attached to a stiff rod (painted red) carried by ants while the robot blade (marked in green) closely follows it. **B** A sketch outlining the experimental setup with the novel cantilever force application mechanism, referred to as the clutch. Key components such as the cargo, the stepper motor, and the position sensor are marked for identification. The experiments are recorded from the top. The angle the cargo makes relative to the nest location is denoted by $\theta$. **C** A zoomed-in snapshot of the clutch mechanism is shown. The blade rotates, engages the clutch, and the vertical screws apply a force on the soft cantilever, causing it to bend. This action transmits forces of around one milinewton to the cargo end. Source data are provided in the Source data file.

However, if the group switches and aligns its direction of motion with the applied force, the switching time ($t_{switch}$) can be determined from the cargo angle time series as the moment when the cargo's angular velocity changes direction. In this scenario, the duration is defined as the time it takes for the group to switch and align with the force, calculated as $\Delta t = (t_{switch} - t_{begin})$. The pulse magnitude was taken as the average calibrated forces magnitude during pulse duration. Pulse durations and magnitudes, as defined here, were used in all subsequent analyses.

The force pulses depicted in Fig. 3B ranged between 0.3 mN and 0.45 mN and lasted for a maximum of 5 s. When compared to the force applied by a single leader ant (-0.1 mN), these forces simulate the joint effort of three to four ants. Some of these pulses reversed the ants' collective motion, while others did not. The stochasticity in the collective response to the external forces is also reflected in our model, as shown in Fig. 3C.

**Response statistics**

We conducted experiments using a wide range of forces ($F_{ext} \in [0.1 \text{ mN}, 1 \text{ mN}]$ over $\Delta t \in [1 \text{ s}, 10 \text{ s}]$). These forces were applied to groups of varying sizes, and their responses were recorded. To ensure the reliability of our study, we performed multiple trials for each group size: $n = 31$ trials for the small group, $n = 85$ trials for the intermediate group, $n = 201$ trials for the large group, and $n = 260$ trials for the largest group. We recorded each group's reaction to the different force pulses, specifically noting possible reversals in the direction of motion.

Figure 4A–D, illustrate the probabilities of direction switching as a function of external force magnitude and duration of the applied force across various group sizes (for the raw experimental data, refer to Fig. S6 in the Supplementary Text). Panels 4E and F display the marginal distributions across all group sizes and demonstrate how switching rates increase with both the magnitude and duration of the external force. Moreover, these plots indicate that larger groups require stronger forces applied over longer durations to switch their direction of motion. Thus, the larger the group, the more resilient it is to the external force[13].

To quantify the prolonged effect of the leader's influence on the group, we introduce a metric called group susceptibility, denoted as $\chi$. This susceptibility measures how sensitive the group is to the forces applied by a newly attached ant. In our experiments, this metric reflects

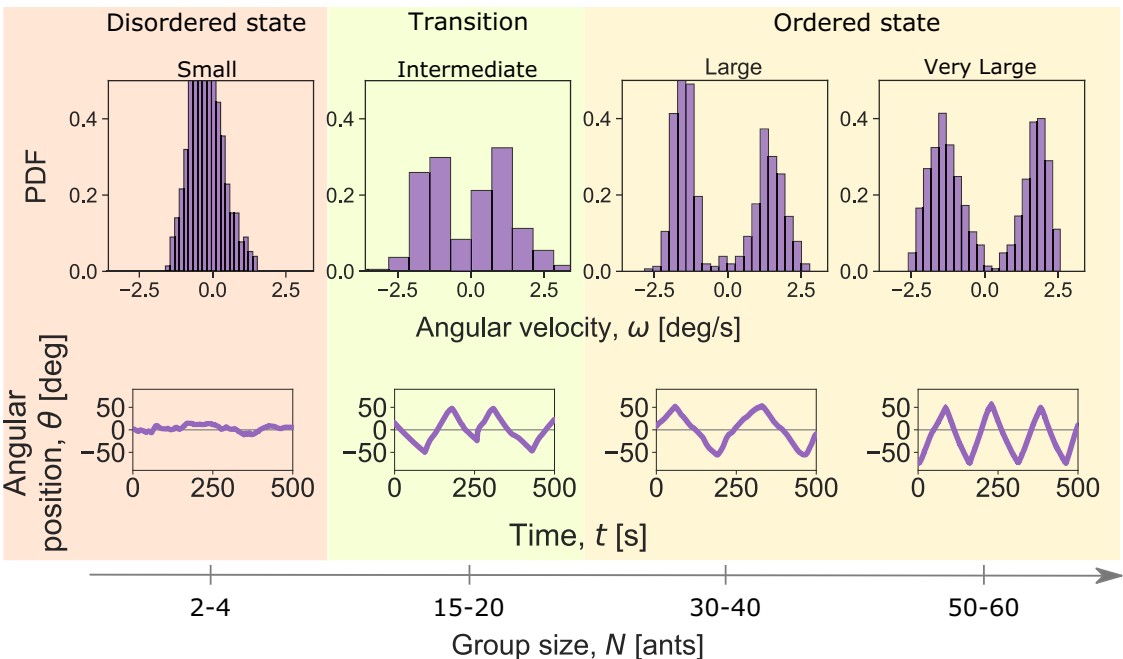

**Fig. 2 | Order-disorder transition during collective transport.** The motion of a hinged cargo as a function of increasing group size in the absence of force application by the robot. The top panels depict angular velocity distributions, while the bottom panels display examples of trajectories. For small group sizes ($N = 2$–$4$ ants), the cargo moves erratically with very low angular velocity and minimal angular change. This behavior results in an unimodal angular velocity distribution centered at zero. In the transition regime, the intermediate-sized cargo with $N = 15$–$20$ ants oscillates with relatively small amplitudes and also exhibits occasional slowing down and direction changes at $\theta = 0°$. In the ordered regime, corresponding to large and very large group sizes ($N = 30$–$40$ and $N = 50$–$60$ ants, respectively), the cargo exhibits persistent oscillations with larger amplitudes, and the angular velocity distributions in this regime are distinctly bimodal. Source data are provided in the Source data file.

how effective the nudge from the robot was not only in switching the group immediately but also in having a long-lasting effect. While this measure of temporal susceptibility was previously proposed theoretically, it had not been directly measured in experiments[10]. Higher group susceptibility indicates that the nudge was strong enough to momentarily disrupt the group's initial alignment and establish a persistent consensus on the new direction. Specifically, we define $\chi$ as the absolute value of the ensemble average of the difference in the cargo's angular position when an external force influences the group compared to when it moves without such influence. Mathematically, this is expressed as $\chi = |\langle \theta(t)_{\text{with force}} \rangle - \langle \theta(t)_{\text{without force}} \rangle|$, where $\theta(t)_{\text{with force}}$ represents the cargo's angular position at a given time $t = t' + \Delta t + \tau$ with the external force, and $\theta(t)_{\text{without force}}$ is the cargo's angular position at time $t = \Delta t + \tau$ after the cargo has crossed the nest location in control cases where no external force was applied. The angle brackets $\langle \cdot \rangle$ indicate the ensemble average. Thus, $\chi$ is calculated by finding the absolute difference between the average angular positions over multiple observations with and without the external force.

We considered the impact of an external force of $0.30 \pm 0.05$ mN, which is equivalent to the effect of 2–3 ants on the group. A force of this magnitude is just sufficient to occasionally disrupt the cohesion within the largest groups. A time delay of $\tau = 5$ s was selected as a sufficiently large time window after the external force has ceased, providing ample time to reliably determine whether the force has a temporary or persistent impact on the group[10]. A higher value of $\chi$ indicates greater susceptibility, signifying an amplified group response to the external force. In Fig. 5, left panel, our findings demonstrate an increase in susceptibility with group size, peaking at the intermediate cargo size before rapidly declining. In the right panel, we present some sample empirical trajectories before and after applying the external force for the three group sizes. The impact of the external force on the group's collective response at a time, $t = t' + \Delta t + \tau$, is clearly evident. For the small group size (A with 2–4 ants), the trajectories of the cargo

exhibit randomness, fluctuating before and after the force application with no consistent consensus. On the other hand, the externally applied force was relatively ineffective for the largest group (C with 50–60 ants) to switch, leading the group to largely disregard the force. In the case of the intermediate size (B with 15–20 ants), while the external force successfully switched the cargo's direction of motion, the group's persistence allowed the cargo to maintain its alignment even after the delay time, $\tau$. These observations align closely with the measured order-disorder transition as manifested by the spontaneous oscillations of the system in the absence of external force, as shown in Fig. 2[10,11]. The transition from disordered to ordered motion occurs around the intermediate cargo size, which coincides with the peak in the system's response, as seen in Fig. 5.

To summarize, during cooperative transport, the participating ants are strongly coupled but can sense the pulling effort from a newly attached ant. The role of an informed puller is to disrupt group alignment and steer the group toward the nest. Whether the collective response is to persistently follow the leader or ignore her will depend on the balance between group size and the influence of the newly attached ants. Any given force in the disordered regime, with the smallest groups of 2 to 4 ants, triggers an immediate response, but one that diminishes quickly. In the ordered regime, with groups of 50 to 60 ants, the same force is insufficient to disrupt the strong persistence of the large group, resulting in minimal to no change in group alignment. Intermediate-sized groups present a unique case. They respond significantly to this newly injected information and exhibit a sustained reaction to the stimulus, characteristic of the maximal response (susceptibility) near a phase transition.

## Theoretical model

In order to understand our empirical findings, we compare them with our theoretical model[10,31]. The model accounts for the stochastic attachment and detachment of ants to the cargo and the single ant

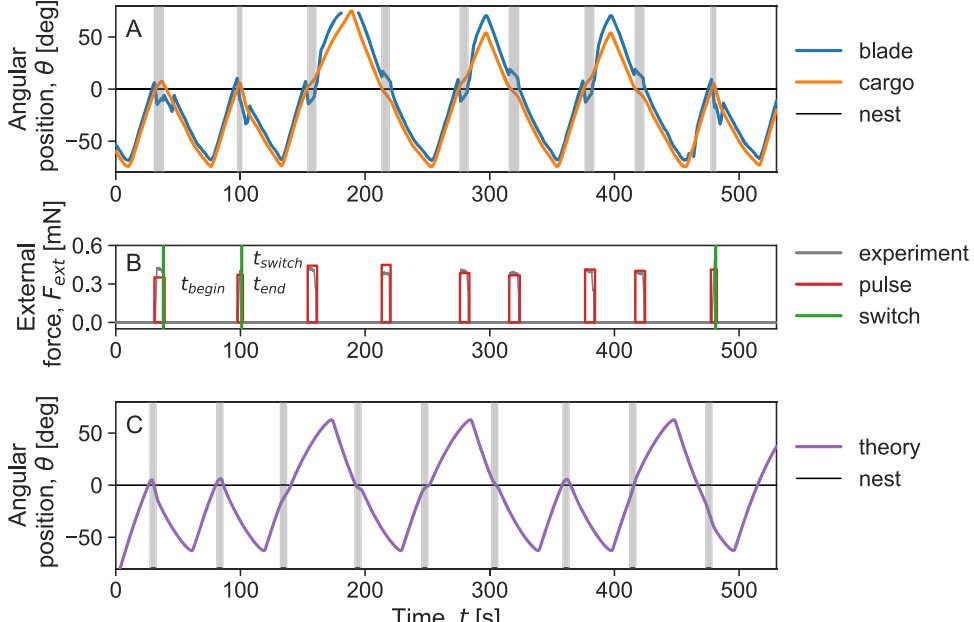

**Fig. 3 | Group response to external forces: experiment and theory. A** A time series of the angular positions of the cargo and robot blade is shown during a sample experimental run. When the two angles match, the robot exerts no force; however, when the blade angle moves away from the load angle (highlighted in gray), an external force is exerted on the ants. The dotted line marks the direction of the nest. In this experimental run, the cargo, carried by a group of 50–60 ants, switched its direction of motion in response to some, but not all, force pulses. **B** The gray lines in the graph represent the angular difference ($\Delta\theta = \theta_{blade} - \theta_{rod}$) between the tracked experimental data for the cargo and the blade after calibration (see, Supplementary Text Fig. S1). A non-zero angular difference indicates the application of a force. The red rectangular pulses in the graph approximate each forcing event. The beginning and end times of these pulses, denoted by $t_{begin}$ and $t_{end}$

respectively, are also shown. If the cargo switched in response to a pulse, the time of the switch, $t_{switch}$, is marked by the green vertical line. **C** Our model is able to replicate both the cargo switching and cargo resisting events that occur when an external force is applied to the cargo, as observed in the experiments. The model parameters used in this case are: $N_{max} = 60$, $F_{ind} = 28$, $f_0 = 2.8$, $k_{on} = 0.021$, $k_{off} = 0.015$, $k_{forget} = 0.09$, $k_c = 1$, $k_{ori} = 0.7$, and $\gamma = 180$ (please refer to *Theoretical model* and *Methods* sections for an explanation of the model parameters). The average number of ants attached to the cargo was $\langle N \rangle = 54$. The forces in the simulation ranged from 0.35–0.45 mN. The forcing events in both the experiments (**A**) and simulations (**C**) are marked by vertical gray bands. Source data are provided in the Source data file.

decisions of whether to pull or lift. These decisions are treated as spin flips in a coupled Ising model. When a new ant attaches, she brings useful directional information to the group and pulls the cargo towards the nest irrespective of the actions of the other ants. In contrast, uninformed ants align their forces with the current direction of motion of the cargo to maximize collective effort. Although lifting ants generally aid the group by reducing friction, in our specific system, the cargo never makes contact with the arena floor, so we simply neglect their effect. The transition rates between puller and lifter, denoted as $r_{p \leftrightarrow l}$ (where $p/l$ denotes an ant in a puller or lifter role), depend on the local force felt by an ant at a specific site $i$. These rates can be expressed as $r_{p \leftrightarrow l} \propto \exp(\mp \mathbf{f}_{loc} \cdot \hat{p}_i / F_{ind})$, where $\mathbf{f}_{loc}$ represents the local force, and $\hat{p}_i$ denotes the ant's orientation as a vector along its body axis, directed from its head along the body axis and indicates the direction of force exertion during pulling. The parameter $F_{ind}$ is inversely related to the coupling strength binding uninformed ants to the group.

The forcing events were incorporated into the theoretical framework using rectangular pulses that matched the magnitudes and durations of the experimentally applied forces, from 0.1 to 1 mN over a duration of 1 to 10 s. We then averaged the group responses over 100 iterations for each pair of $(F_{ext}, \Delta t)$. If the group switches and aligns its direction of motion with the applied force, the switching time ($t_{switch}$) can be identified when the cargo's angular velocity reverses direction. The duration is defined as the time taken for the group to switch and align with the force, calculated as in the experiments (see previous section).

In addition to the "microscopic" simulations, we use the mean-field equations (see derivation in the *Materials and Methods*) to obtain an approximate analytical expression that describes the relationship

between the magnitude of the external force ($F_{ext}$) and the duration of the applied forcing ($\Delta t$) that are needed to prompt the cargo to switch its direction of motion[11]. We derive the expression: $F_{ext} = \gamma v_0 \beta / (1 - \exp(-2k_c \Delta t \beta))$, see Eqn. (13). In this equation, $v_0$ and $\gamma$ represent the cargo speed and damping for specific group size, $k_c$ denotes the rate of role switching among agents, and $\beta = f_0 N / 2F_{ind} - 1$. We find that this expression fits the 50% likelihood of switching probability per time bin both in the empirical data presented in Fig. 4A–D and in the simulations of the microscopic model shown in Fig. 4G–J. Although the derivation of the mean-field expression assumes $v \sim 0$, surprisingly, it seems to work well even at higher speeds for the larger group sizes. Moreover, in Fig. 4K and L, we plot the mean switching probability per time bin as a function of the force duration ($\Delta t$), independent of the force magnitude ($F_{ext}$), and as a function of the force magnitude, independent of its duration, respectively. Our findings indicate that as group size increases, the transition region shifts toward stronger forces and longer durations, which is in agreement with the mean-field relation. Overall, we find good agreement between the empirical data and the theoretical simulations. Thus, these observations support our conclusion that the combinations of force and duration that reverse the cargo's direction of motion are consistent with our model based on tactile interactions among the carrying ants. Any discrepancies may stem from the inherently stochastic nature of the experiments and individual differences among ants.

To measure the susceptibility of the ant group to the external force using the theoretical model, we apply in our simulations force pulses ranging from 0.25 mN to 0.35 mN to groups of ants ranging from the smallest (two ants) to the largest (70 ants) and average the numerical results over 100 iterations for each group size. By taking the

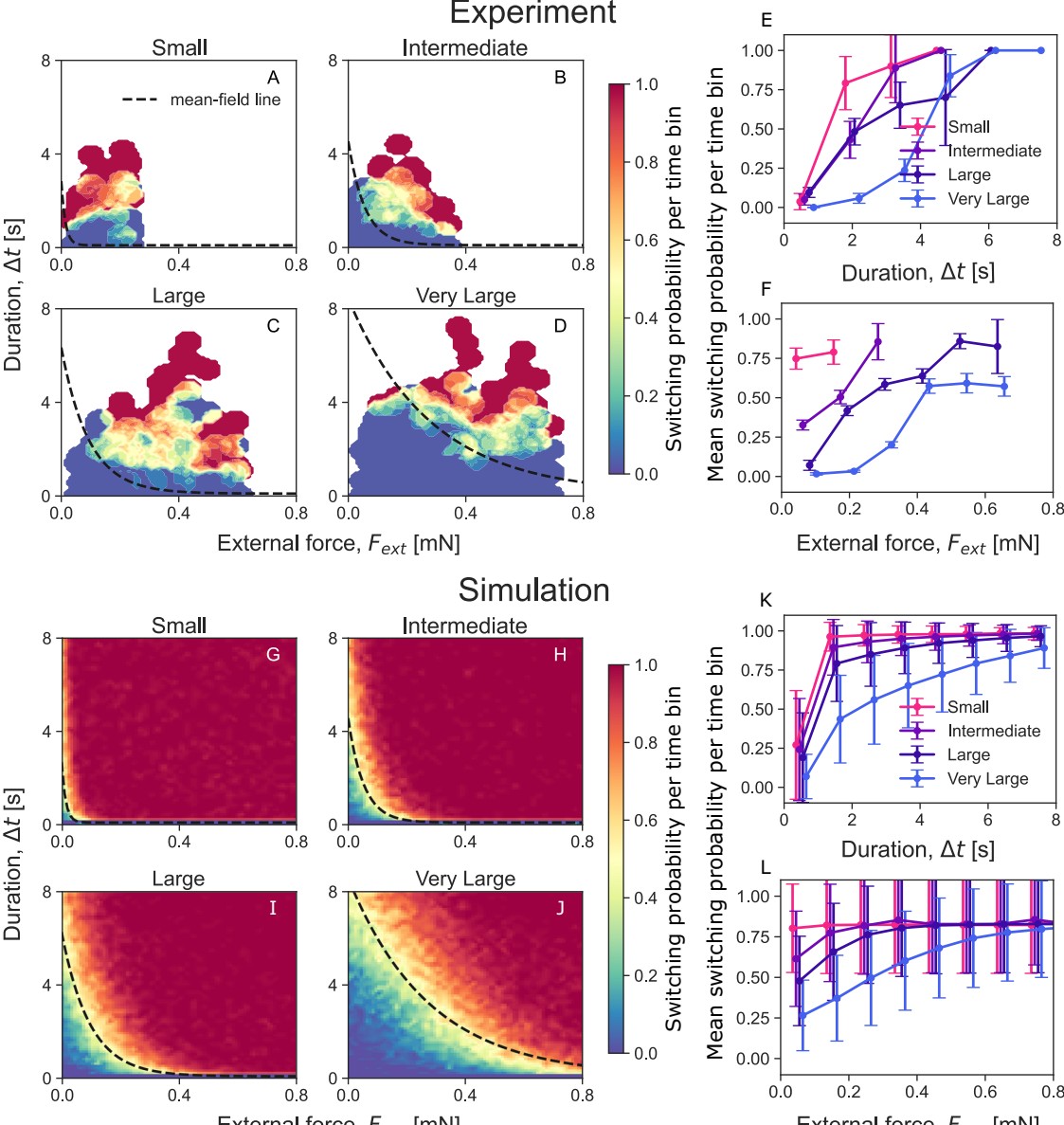

**Fig. 4 | Collective response to external forces.** In (**A**–**D**), the heatmap illustrates the switching probability per time bin derived from empirical data, plotted as a function of both the duration ($\Delta t$) and magnitude of the externally applied force ($F_{ext}$). The data has been binned using $k$-nearest neighbors and averaged through a simple moving window across all group sizes (the raw empirical data is presented as scatter points in Fig. S6 of the Supplementary Text). The empirical data consists of multiple trials for each group size: (**A**) $n = 31$ for small, (**B**) $n = 85$ for intermediate, (**C**) $n = 201$ for large, and (**D**) $n = 260$ for very large groups. **G**–**J** Present the results from simulations of the microscopic model conducted over a $100 \times 100$ grid of $F_{ext}$ and $\Delta t$, averaged across 100 runs for each parameter pair as heatmaps. The color scale in both heatmap panels ranges from dark red, indicating a high probability of

group switching, to dark blue, indicating resistance to switching. The phase boundary corresponds to the yellowish-green area in the heatmaps, where the probability of switching is 50%. It is fitted with exponential functions from Eqn. (13), shown as dashed lines. **E**, **K** Show the mean switching probability per time bin as a function of the magnitude of the externally applied force ($F_{ext}$), independent of the duration of the force from empirical data and numerical simulations, respectively. **F**, **L** Show the mean switching probability per time bin as a function of the forcing duration ($\Delta t$), independent of the magnitude of the external force from empirical data and numerical simulations, respectively. The error bars represent the standard deviations from the mean values. Source data are provided in the Source data file.

ensemble-averaged angular position of the cargo at time $t = \tau$ s after the force is applied at $t = t'$ s over $\Delta t$ s and comparing it to the ensemble-averaged angular position of the cargo at the same time at $t = t' + \Delta t + \tau$ s in the absence of any external force, we obtain susceptibility, $\chi = |\langle \theta(t)_{\text{with force}} \rangle - \langle \theta(t)_{\text{without force}} \rangle|$ for each group size to a fixed external force. Similar to the susceptibility calculations in experiments, we also choose a delay time window of $\tau$ of 5 s in the numerical simulations, which allows for transient dynamics to settle after the force application, ensuring that the measured susceptibility reflects the persistent effect of the force rather than temporary fluctuations. In

Fig. 5, left panel, we present the results of numerical simulations for the susceptibility measure. In the simulations, susceptibility peaks at the intermediate group size of 20 ants, in agreement with the experiments where the maximal group response is observed for 18–24 ants, corresponding to the intermediate cargo size[10,31].

To gain intuition on cargo switching dynamics in response to the external force, we explore the microscopic behavior of the agents in the simulations. As the cargo moves along the constrained circular path defined by the stiff rod, the ants at the front take on the role of pullers, while the ants at the back act as lifters. The simulation

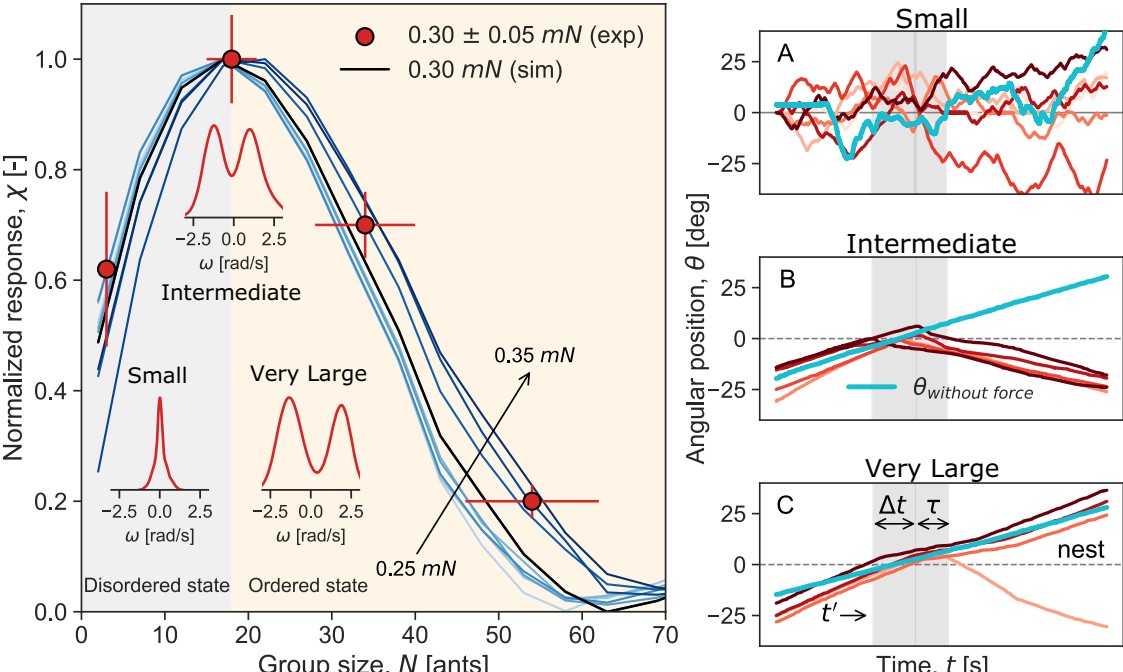

**Fig. 5 | Peak susceptibility at a critical group size.** In the left panel, scatter points show the group response to external forces in experiments for $F_{ext}$ in the $0.30 \pm 0.05$ mN range for the four cargo sizes (with diameters of 0.3, 1, 2, and 4 cm) and average number of ants, $N = 3$ for the small cargo, $N = 18$ ants for the intermediate cargo, $N = 34$ ants for the large cargo, and $N = 54$ ants for the largest cargo. The error bars are the standard error of the mean. The continuous curves are the normalized response curves obtained from the numerical simulations for group sizes $N = 2$–70, averaged over 100 runs each for a range of forces between 0.25 and 0.35 mN. The response curve for $F_{ext} = 0.30$ mN is the dashed black curve in the figure. The inset plots show the angular velocity distribution of the cargo in the disordered state (cargo size: small), at the transition zone (cargo size: intermediate), and in the ordered state (cargo size: very large), all empirically derived from experiments in the absence of any external force from the robot (also shown in Fig. 2). On the right, snapshots of cargo trajectories from experiments are displayed for the three group sizes before, during, and after the application of the external force. The trajectories exhibit fluctuations before and after for the smallest group size (**A**), suggesting no consensus or persistence. For the intermediate group size (**B**), the group demonstrates consensus by switching its response to the external force and persisting after the delay time has elapsed. In contrast, the largest group (**C**) resists the force and continues moving along its original direction of motion. The time stamps, denoted as $t'$, represent the time of force application, while $\Delta t$ signifies the duration of force application. Additionally, $\tau$ denotes the delay time for measuring group persistence. A fixed time delay of $\tau = 5$ s is utilized in both experiments and simulations, and the maximal response is used to normalize values in both cases. A particular instance of the angular position of the cargo, $\theta(t)_{without\ force}$, in the absence of the external force, is shown in cyan. The angular data for all cargo sizes has been rotated to reflect that the external force always points in the same direction. Source data are provided in the Source data file.

snapshots in Fig. 6 illustrate the behavior of the cargo and the ants attached to the cargo before, during, and after applying an external force. In Fig. 6A, an external force of 0.4 mN is applied to the cargo at time $t = 0$ s. This force is strong enough to switch the cargo's initial direction of motion, as shown in Fig. 6B. Figure 6C shows the total ant force (in the $x$-direction), where the external force counteracts the ants' pulling force. The ant force gradually decreases to zero between $t = 0$ s and $t = 4$ s, and the cargo is stalled. In this interval, the number of puller ants attached to the front of the cargo rapidly decreases, and the force is just sufficient to trigger role-switching among the lifters at the back of the cargo by turning them into pullers as seen from Fig. 6D. With enough pullers at the back, that now outweighs the number of pullers at the front. the cargo switches its direction of motion.

By contrast, in Fig. 6E, a weaker force of 0.2 mN is applied. In this case, the cargo resists the external force and maintains its original direction of motion. Between $t = 0$ s and $t = 4$ s, the external force successfully stalls the cargo by counteracting the total ant force as seen in Fig. 6F, G. However, in this case, the external cannot initiate sufficient role-switching among the lifters at the back to convert enough of them to pullers. Moreover, the number of pullers at the back remains unchanged from $t = 4$ s to $t = 8$ s as shown in Fig. 6H. After $t = 8$ s, the number of puller ants at the front exceeds the number of pullers at the back, allowing the cargo to continue its original direction of motion.

The microscopic model provides important insights into the switching dynamics of the cargo. In both cases shown in Fig. 6, the

external force slows down the cargo and can even stall it entirely. However, whether the cargo switches direction or persists in the same direction is dictated by the response of the individual ants to the force they feel and whether they maintain or switch their role. If the external force is just sufficient to slow down the cargo and stall it, $F_{ant} \simeq F_{ext}$, then the cargo's subsequent direction of motion after the force ceases is random and depends on the instantaneous number of pullers on one side versus the other. When the force is strong enough such that $F_{tot} = F_{ant} - F_{ext} < 0$, it is capable of switching the roles of a sufficient number of puller ants, triggering a cascade of switches among them. Due to the strong coupling between ants and their tendency to follow the behavior of other ants, this cascade leads to a collective shift at the group level. With sufficient pullers at the back, the cargo will be forced to change its original direction of motion. More details about the switching rates between pullers and lifters can be found in the Supplementary Text.

## Discussion

During cooperative transport, all ants are attached to the same load and, thus, mechanically linked. In this all-to-all interaction, the coupling constant scales linearly with group size[31]. Therefore, varying group sizes provide a simple means by which the system can be transitioned between ordered and non-ordered regimes. This unique dependence on system size offers a rare opportunity for experimentally measuring how group susceptibility varies near the transition

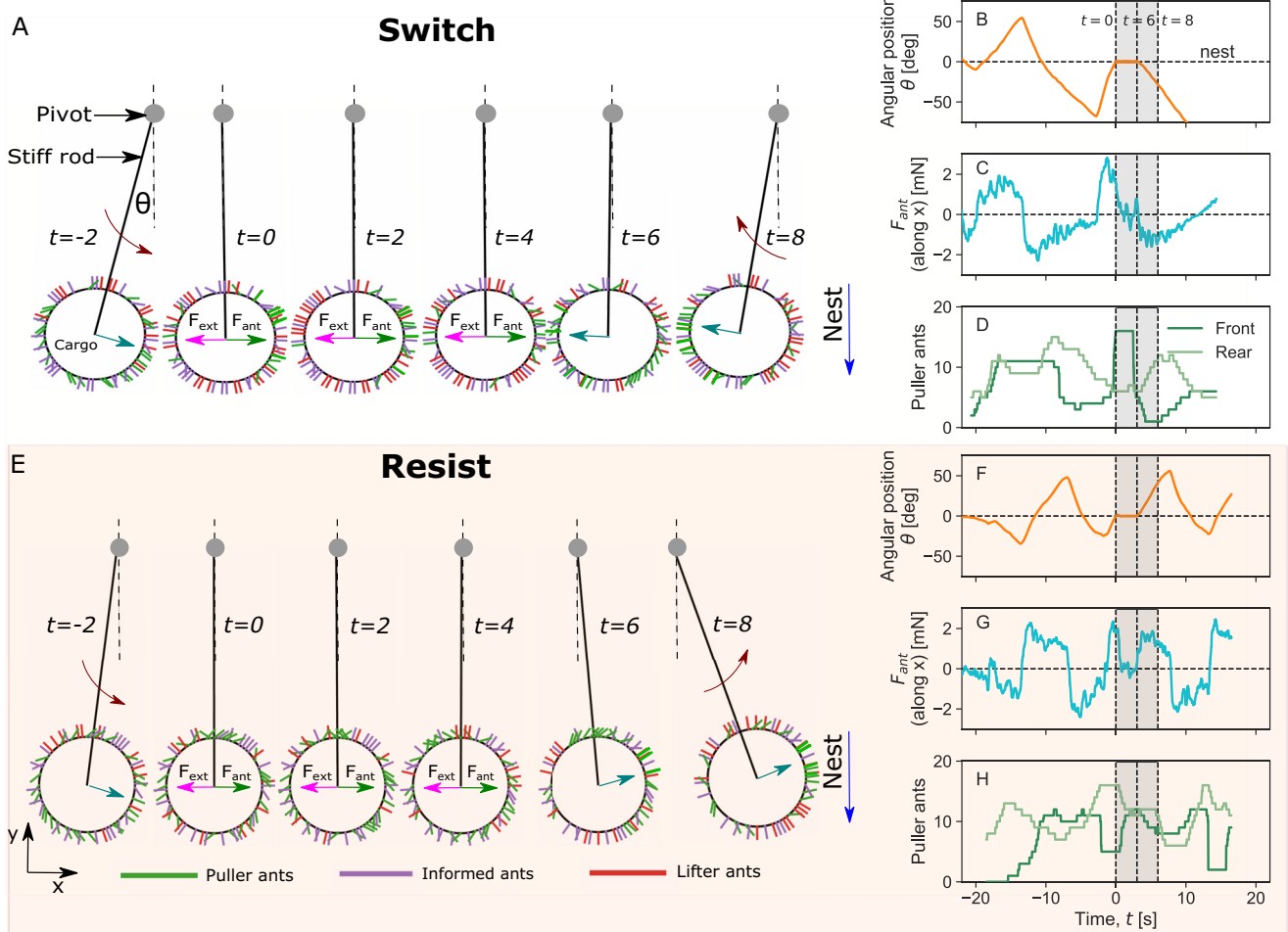

**Fig. 6 | Simulation snapshots of the cargo before and after the external force.** The figure shows the ants' cargo dynamics and microscopic behavior in response to external forces for the largest group size with $N_{max} = 60$ ants. **A–D** Show the cargo's response to a strong external force of 0.4 mN, resulting in a switch in its original direction of motion. The simulation snapshots of the cargo with ants attached are shown in (**A**). In (**B**), the angular position of the cargo is shown as a function of time. **C** Shows the total force exerted by the ants in the $x$-direction. **D** Shows the number of puller ants attached to the front and back of the cargo. **E–H** Show the cargo's response to a weaker external force of 0.2 mN. In this case, the cargo resists the external force and continues along its initial direction of motion, as seen in the simulation snapshots in (**E**). In (**F**), the angular position of the cargo is shown as a function of time. The total force the ants exert in the $x$-direction is shown in (**G**), while in (**H**), the number of puller ants attached to the front and back of the cargo is shown. The external force is introduced at time $t = 0$ s, acting in the opposite direction of the cargo's angular position. The vertical dashed lines in (**B–D**, **F–H**) denote the time stamps corresponding to the simulation snapshots shown in (**A**, **E**). Source data are provided in the Source data file.

point. To realize this opportunity, we needed a method to apply controlled forces to a mobile object. Our technique involved motion around an immobile hinge, effectively allowing a stationary robot to couple with the mobile ants. The robot can exert ant-scale millinewton forces that emulate a transient leader and perturb the group in real-time, thus allowing us to measure group response and quantify susceptibility.

As opposed to a natural setting where a single ant with directional information can influence the group to make a series of smaller turns, a relatively stronger force equivalent to that of 2–3 ants is needed to change the cargo's direction of motion in the experiments, that necessitates a 180° switch. By gently nudging the cargo toward the nest, the robot assumes the role of an informed puller and subtly injects useful directional information into the group. Figure 4 illustrates that in both experiments and numerical simulations, the introduction of directional information by the robot can sometimes disrupt group cohesion, compelling the ants to switch directions. Conversely, there are scenarios where the strong cohesion in the group makes it impervious to this new information. The impact of this newly introduced information depends on several factors, and variability may stem from changes in group size

caused by random attachments and detachments to and from the cargo and individual differences among the ants.

When longhorn crazy ants engage in cooperative transport, each ant either pulls on the load or lifts it. These binary choices can be described using a spin-like (Ising-type) model. The fact that all ants are connected and influence each other through a shared cargo, effectively translates into all-to-all interactions. The resulting model, in its thermodynamic limit, predicts an order-disorder phase transition as a function of the coupling parameter, $F_{ind}$. We have further shown that the experimentally modifiable group size can serve as a control parameter instead of the coupling strength[10]. We theoretically demonstrated that the ant collective indeed undergoes an order-to-disorder transition as a function of group size[10,11]. In agreement, the motion of the load carried by ants switches from ballistic motion to a biased random walk at the predicted group size[31]. The thermodynamic limit of the model further tells us that at this critical transition point, the susceptibility to small external fields diverges. Experimentally, we can only measure short-time responses, specifically, how much the ant group moves in response to an applied force. Our analytical mean-field solution and numerical simulations of the Ising model on finite sizes

predict that, in this case, a peak in responsiveness will be evident at the phase transition (critical group size). As reported in this work, this peak is indeed observable in a group of carrying ants. Therefore, although criticality and divergence of susceptibility are usually defined in the thermodynamic limit of large systems, our empirical results, together with our microscopically grounded model, establish a clear link between the phase transition that the model predicts in the thermodynamic limit and the finite-size order-disorder transition observed in real ant groups.

The observation that the collective response peaks at an intermediate group size during cooperative transport in *P. longicornis* ants, supported by both empirical evidence and theoretical analysis, raise fundamental questions about their biological significance and relevance. This increased susceptibility to minimal perturbations—at the scale of a single "leader" ant—enables the group to be maximally responsive while collectively transporting food in their natural environment. Notably, this enhanced responsiveness is observed only when the group size is approximately 20 ants – the typical group size seen in *P. longicornis* during cooperative transport in their natural habitat. Unlike other species, *P. longicornis* ants exhibit a balance between conformism and individuality, facilitating a more distributive approach to decision-making during cooperative transport[31]. This balance ensures that the success of the group relies on the influence of one or few newly attached individuals. These influential individuals cannot move the large cargo alone, and the group serves to amplify their knowledge into a force that enables efficient navigation of food to the nest. This idea has been proposed since the microscopic model was formulated[10], and here we have provided the first direct empirical demonstration of this maximal responsiveness to an externally applied "robot-ant" force. The heightened susceptibility implies that the ant collective is in the critical regime near the transition between collective states, where it is most responsive to the new information injected by a single leader ant and can rapidly adapt to this new information. This state of criticality ensures that useful information, such as the direction to the nest, is disseminated efficiently throughout the group, thereby enhancing the collective ability to maneuver obstacles and transport effectively. Thus, being close to a transition regime that exhibits finite-size critical behavior has practical advantages, such as enhancing efficiency and adaptability in performing collective tasks.

In conclusion, our study bridges the gap between theoretical predictions and empirical observations by directly measuring susceptibility and its peak at an intermediate group size in *P. longicornis* ants. Our findings validate the criticality hypothesis in biological collectives. Thus, this work not only advances our understanding of collective behavior in ants but also contributes to the broader discourse on the role of criticality in biological collectives.

## Methods

### Field experiments
Data was collected from two nests of *P. longicornis* ants in the Weizmann Institute of Science in Israel. The experiments took place during the summer when these ants display cooperative transport behavior. An arena measuring $60 \times 55$ cm was used for the ants to transport cargoes of different sizes collectively. The testing arena was strategically positioned to ensure optimal filming conditions, with a flat floor and uniform illumination, and located about 1 m from the nest. This setup provided a controlled environment in their natural habitat for observation. The cargo comprised specially designed silicone rubber rings, each 1.5 mm thick and with diameters 0.3, 1, 2, and 4 cm. The rings were designed with edges suitable for the ants to grasp with their mandibles during cooperative transport. The cargoes were made appealing for the ants as they were coated with cat food from either the Royal Canin™/Happy Cat™ brands. Each cargo was affixed to one end of a rigid rod, allowing it to rotate about the other end freely. A rotary encoder at this fixed axis detected the shaft's rotation and recorded its angular position. A gap of ~0.5 cm was maintained between the cargo and the experiment board to eliminate friction and drag during experiments. High-definition recordings of the experiments were captured at 25 frames per second (fps) using a Panasonic HC-VX870 camcorder mounted on a stand. The experiments took place daily in a clean arena, with the cargo freshly coated with cat food before each session. The first 10–20 min involved the recruitment phase to obtain a steady influx of ants. Once we observed sustained oscillations in the cargo, we introduced forcing events, which involved short nudges performed by a robot programmed to execute predefined combinations of force and duration. Each nudge was kept short relative to the natural oscillation period of the cargo. Applying force over a longer interval would significantly change the cargo angle during the push, thereby altering the system conditions mid-measurement and obscuring our ability to observe the short-time response under a fixed set of parameters. The short-duration forcing thus allows us to measure how the ant group responds under near-instantaneous, controlled perturbations, rather than confounding that response with the changing rod position.

### Movie processing
A dedicated image-processing program was implemented in Python 3.10.12 with OpenCV 4.9.0 for feature recognition, while data handling and visualization used NumPy 1.26.4 and Matplotlib 3.8.4. The program accurately recognizes the center of the blade (marked with a green sticker) and the centerline of the stiff rod (painted red) attached to the cargo. By selecting three points along the circular path of the cargo's center of mass, the program determines the common center of rotation for both the cargo and the blade. As a result, the program captures and stores the positional data of the blade $\left(\theta_{\text{blade}}\right)$ and the cargo $\left(\theta_{\text{cargo}}\right)$ frame by frame. The angular velocity is calculated numerically using the central difference method, taking into consideration the frames before and after the current frame: $\omega = ((\theta(t + \Delta t) - \theta(t - \Delta t))/2\Delta t) \times \text{fps}$. The force application instances were derived from the tracked time series data of the cargo and the blade as they passed by the nest location. Each force application event involved grouping the timestamps for the start, end, and switch in the blade and cargo time series data using *k*-means clustering.

### Data analysis
For a given magnitude of external force, $F_{\text{ext}}$, a group may either resist the force or switch its alignment in response. To determine the switching probability per time bin, the entire time range—from 0 to the maximum duration of force application observed across all trials for that force – is first divided into evenly spaced bins. The duration of force application in each trial is then mapped onto these bins. In a particular trial, for a given external force magnitude, if the cargo does not switch throughout the entire duration $\left(\Delta t = t_{\text{end}} - t_{\text{begin}}\right)$, then all corresponding bins are recorded as "no switch" bins. However if the cargo does switch its direction of motion at time $t_{\text{switch}}$, we record all bins prior to that moment as "no switch", and the bin in which the switch occurs $\left(\Delta t = t_{\text{switch}} - t_{\text{begin}}\right)$ is marked as a "switch bin". The switching probability per time bin for each group size is then represented as heatmaps (shown in Fig. 4), generated by binning the data using a *k*-nearest neighbor approach and smoothing with a simple moving average.

### Microscopic model
The empirical observations are supported by theory through numerical simulations that describe the entire stochastic dynamic of the ant-cargo system under the influence of an external force. The cargo ring carried by the ants is modeled as a circle with equally spaced sites labeled by an angle $\alpha_i, i \in [1, N_{\text{max}}]$ where $N_{\text{max}}$ represents the maximum number of sites on the cargo. Each site can either be empty or occupied by a puller or a lifter. Surrounding ants attach

to the cargo at a constant rate $k_{on}$. A newly attached ant is considered an informed puller, and she tries to steer the cargo towards the nest, $\theta = 0°$ in the simulation. An informed ant can transition to an uninformed puller at $k_{forget}$, while attached ants can detach from the cargo at $k_{off}$. Informed ants pull in the direction of the nest, while uninformed puller ants pull in the direction of the force they sense. The force by a puller ant at a cargo site, $i$, is expressed as $\mathbf{f}_i = f_0 \hat{p}_i$ where $\hat{p}_i = \cos(\alpha_i \pm \phi_i) + \sin(\alpha_i \pm \phi_i)$ represents the body axis vector of the ant. Here, the angle $\phi$ denotes the orientation of the ant about the outward site normal and is constrained by $\phi_{max} = 52°$. Transition rates between pulling and lifting ($p \leftrightarrow l$) are dictated by, $r_{p \to l} = k_c \exp(-\mathbf{f}_{loc} \cdot \hat{p}_i / F_{ind})$ and $r_{l \to p} = k_c \exp(\mathbf{f}_{loc} \cdot \hat{p}_i / F_{ind})$, where $k_c$ denotes the basal conversion rate and $\mathbf{f}_{loc}$ represents the local force felt by an ant at site $i$. Since the system is rigid and does not involve rotations, the local force $\mathbf{f}_{loc}$ is uniform for all ants and is equivalent to the force at the center of mass of the cargo. The time until the next stochastic event, $\Delta t$, is obtained by drawing a random number from a uniform distribution, $\mathcal{U}(0, 1)$ such that $\Delta t = -1/R_{total} \log(\text{rand})$, where $R_{total}$ represents the total rate of all possible events. The simulation progresses in discrete time steps of dt = 0.01 s, incrementally updating the state of the systems, thus allowing for the continuous approximation of the evolution of the system over time. Each step incrementally advances the simulation until $\Delta t$ is reached, at which point, the next stochastic event, determined by the Gillespie algorithm, is executed[10,12]. When the cargo points toward the nest, located at ($\theta = 0°$), an external force, $\mathbf{F}_{ext}$, is applied. The model has numerous parameters, including the maximum number of sites for ants to attach ($N_{max}$), damping coefficient ($\gamma$), and several variables representing basal rates of attachments ($k_{on}$), detachments ($k_{off}$), forgetting ($k_{forget}$), and decision-making ($k_c$). In our simulations, we kept the rates $k_{off}$, $k_{for}$, $k_c$, and $k_{ori}$ unchanged from previous studies[10]. However, we increased the value of $k_{on}$ to ensure that the cargo is always fully saturated with ants, mirroring our experimental observations. To account for the intermittent "jerks" that the agents (ants) experience when the robot nudges the cargo in experiments, we also increased the individuality parameter $F_{ind}$ to 28, compared to the value of 10 used in earlier works[10]. The maximum number of attachment sites $N_{max}$ varies with each group, and the damping coefficient $\gamma$ scales linearly with $N_{max}$. Finally, we calibrated the model using the angular velocity and oscillation amplitude measured from the experiments for each cargo. For a comprehensive overview of the model, force application, and parameter estimation, please see the Supplementary Text.

## Mean-field approximation

In the limit when the cargo is fully saturated with ants, each ant can detect the collective force exerted by all the others, making the microscopic model mean field. An analytical expression for the mean-field ant-cargo system can be written to characterize the mechanics of the system. When subjected to the external force, $F(t)$ the total force experienced by the cargo is given by,

$$F_{tot} = f_0 n_p^{front} - f_0 n_p^{rear} - f_0 G \sin \theta - F(t) \qquad (1)$$

Here, $f_0$ represents the force exerted by a single ant, $G$ denotes the number of informed ants, $n_p$ signifies the number of ants pulling, and $\theta$ is the cargo angle measured relative to the nest, located at $\theta = 0°$. The amount of force exerted depends on the number of individuals pulling on each side of the cargo. The following equations express the rate of change in the number of pullers on either side of the cargo:

$$\left(\frac{dn_p}{dt}\right)^{front} = \left(r_{l \to p} n_l - r_{p \to l} n_p\right)^{front} \quad \text{and} \quad \left(\frac{dn_p}{dt}\right)^{rear} = \left(r_{l \to p} n_l - r_{p \to l} n_p\right)^{rear} \qquad (2)$$

The transition rates between pulling and lifting are determined by[11],

$$\left(r_{p \to l}\right)^{front/rear} = k_c \exp\left(\mp \frac{F_{tot}}{F_{ind}}\right) \quad \text{and} \quad \left(r_{l \to p}\right)^{front/rear} = k_c \exp\left(\pm \frac{F_{tot}}{F_{ind}}\right) \qquad (3)$$

Given that the ant-cargo system is over-damped, the total force $F_{tot}$ equals $\gamma v$, where $\gamma$ represents the damping coefficient. By taking the time derivative of the total force $F_{tot}$ and solving the equation of motion, we can determine,

$$\frac{d\theta}{dt} = \frac{v}{l_{rod}} \qquad (4)$$

$$\frac{1}{k_c} \frac{dv}{dt} = \frac{f_0 N}{\gamma} \sinh\left(\frac{v}{F_{ind}/\gamma}\right) - 2\left(v + \frac{f_0 G}{\gamma} \sin \theta + \frac{F(t)}{\gamma}\right) \cosh\left(\frac{v}{F_{ind}/\gamma}\right)$$
$$- \frac{f_0 G}{\gamma} \frac{1}{k_c} \frac{v \cos \theta}{l_{rod}} - \frac{1}{\gamma} \frac{dF(t)}{dt} \qquad (5)$$

The external force from the robot, $F(t)$, is implemented as a rectangular pulse at $t = t'$ over a duration $\Delta t$ when the cargo points toward the nest. Thus, $F(t) = F_{ext}$ for $t' \leq t \leq t' + \Delta t$, and 0 otherwise. As a result, the velocity equation in Eqn. (5) can be expressed using Heaviside and Delta function notation as shown below,

$$\frac{1}{k_c} \frac{dv}{dt} = \frac{f_0 N}{\gamma} \sinh\left(\frac{v}{F_{ind}/\gamma}\right) - 2\left(v + \frac{f_0 G}{\gamma} \sin \theta\right) \cosh\left(\frac{v}{F_{ind}/\gamma}\right) - \frac{f_0 G}{\gamma} \frac{1}{k_c} \frac{v \cos \theta}{l_{rod}}$$
$$- \frac{2F_{ext}}{\gamma} \cosh\left(\frac{v}{F_{ind}}\right) (H(t - t') - H(t - (t' + \Delta t)))$$
$$- \frac{F_{ext}}{\gamma} (\delta(t - t') - \delta(t - (t' + \Delta t))) \qquad (6)$$

Please refer to the Supplementary Text for the full derivation of the ant-cargo pendulum under mean-field approximation.

## Force response curve

We seek to derive an approximate analytical expression for the relationship between the magnitude of the applied force $F_{ext}$ and the duration time of the applied force $\Delta t$, which leads to switching in the direction of motion. Our aim is to characterize the boundary illustrated in Fig. 4 and Fig. S6 in the Supplementary Text, where the applied force and its duration are sufficient to induce direction of motion switching.

When the length of the rod, $l_{rod}$, significantly exceeds 1, Eqn. (6) can be simplified as follows:

$$\frac{1}{k_c} \frac{dv}{dt} = \tilde{N} \sinh\left(\frac{v}{\tilde{F}_{ind}}\right) - 2v \cosh\left(\frac{v}{\tilde{F}_{ind}}\right)$$
$$- \frac{2F_{ext}}{\gamma} \cosh\left(\frac{v}{\tilde{F}_{ind}}\right) (H(t - t') - H(t - (t' + \Delta t))) \qquad (7)$$
$$- \frac{F_{ext}}{\gamma} (\delta(t - t') - \delta(t - (t' + \Delta t)))$$

Before the switch ($t' \leq t < t' + \Delta t$), the Heaviside function $H(t - t')$ equals 1 because $t > t'$, and $H(t - (t' + \Delta t))$ equals 0 because $t < t' + \Delta t$. The delta functions do not contribute because $t$ does not equal $t'$ or $(t' + \Delta t)$. The differential equation simplifies to:

$$\frac{1}{k_c} \frac{dv}{dt} = \tilde{N} \sinh\left(\frac{v}{\tilde{F}_{ind}}\right) - 2v \cosh\left(\frac{v}{\tilde{F}_{ind}}\right) - \frac{2F_{ext}}{\gamma} \cosh\left(\frac{v}{\tilde{F}_{ind}}\right) \qquad (8)$$

We utilize the small angle approximation for the hyperbolic trigonometric functions, approximating $\sinh x$ as $\approx x$ and $\cosh x$ as $\approx 1$. With this, we can express the equation as follows:

$$\frac{1}{k_c}\frac{dv}{dt} = \left(\frac{\tilde{N}}{\tilde{F}_{\text{ind}}} - 2\right)v - \frac{2F_{\text{ext}}}{\gamma} \tag{9}$$

The cargo initially moves with a velocity of $v_0$. At time $t = t'$, when the cargo is pointing toward the nest, an external force is applied over a duration of $\Delta t$. We assume that the force is strong enough to cause the cargo to reverse its original direction of motion after time $t = t' + \Delta t$ has elapsed. As a result, at exactly $t = t' + \Delta t$, the cargo's velocity momentarily becomes 0. Therefore, we can integrate the above equation from $t = t'$ to $t = t' + \Delta t$ and from $v = v_0$ to $v = 0$:

$$\int_{v_0}^{0} \frac{dv}{\left(\frac{\tilde{N}}{\tilde{F}_{\text{ind}}} - 2\right)v - \frac{2F_{\text{ext}}}{\gamma}} = k_c \int_{t'}^{t'+\Delta t} dt \tag{10}$$

After evaluating the limits, we arrive at the following expression:

$$\ln\left|\frac{B}{Av_0 + B}\right| = Ak_c\Delta t, \quad \text{where} \quad A = \frac{\tilde{N}}{\tilde{F}_{\text{ind}}} - 2 \quad \text{and} \quad B = -\frac{2F_{\text{ext}}}{\gamma} \tag{11}$$

Rearranging the terms, and writing $F_{\text{ext}}$ as a function of $\Delta t$ yields the following final form:

$$F_{\text{ext}} = \gamma v_0 \left(\frac{\tilde{N}}{2\tilde{F}_{\text{ind}}} - 1\right)\left(\frac{1}{1 - \exp\left(-2\left(\frac{\tilde{N}}{2\tilde{F}_{\text{ind}}} - 1\right)k_c\Delta t\right)}\right) \tag{12}$$

Considering that $\tilde{N} = f_0 N/\gamma$ and $\tilde{F}_{\text{ind}} = F_{\text{ind}}/\gamma$, the expression $\left(\tilde{N}/2\tilde{F}_{\text{ind}} - 1\right)$ simplifies to $\left(f_0 N/2F_{\text{ind}} - 1\right)$. We substitute $\left(f_0 N/2F_{\text{ind}} - 1\right)$ by $\beta$ and rewrite Eqn. (12),

$$F_{\text{ext}} = \frac{\gamma v_0 \beta}{1 - \exp\left(-2\beta k_c\Delta t\right)} \tag{13}$$

The parameter $\beta$ is explicitly dependent on $N$ and allows us to span a range of regimes, from the disordered phase to the ordered phase. When $N$ is small, $\beta$ is negative, indicating that the system is in a disordered state. On the other hand, when $N$ is large, $\beta$ is positive, signifying that the system is in a ordered phase. We use Eqn. (13) to obtain the mean-field transition lines shown in Fig. 4A–D and G–J. To fit the analytical function, only those pairs of $(F_{\text{ext}}, \Delta t)$ are chosen for which the switching probability per time bin is 50%. While the parameters $v_0$, $\gamma$, $\beta$, and $N$ vary with group size, the value of $k_c = 0.7$ remains constant across the four group sizes. Thus, a least-square fitting is performed over the independent variable $\Delta t$ for a given group size and the corresponding model parameters to obtain the best fit for $F_{\text{ext}}$.

## Data availability
The datasets used to generate the figures are provided in the Supplementary Files. Source data are provided with this paper.

## Code availability
All tracking and simulation scripts are openly accessible at https://github.com/ant-lab-hub/robot-leader and a citable version of the version used in this study has been also uploaded to Zenodo[43].

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

## Acknowledgements

O.F. acknowledges support from the European Research Council (ERC) under the European Union's Horizon 2020 research and innovation program (Grant Agreement No. 770964). We thank Efi Efrati for his help conceiving the novel force-application mechanism; Guy Han and Yuri Burnishev for technical support; and Ehud Fonio for help with the initial experiments. We also appreciate the helpful comments on the first manuscript draft from other Antlab members, including Tabea Dreyer, Lior Baltiansky, Harikrishnan Rajendran, and Xiahui Guo. O.F. is the incumbent of the Henry J. Leir Professorial Chair, and N.S.G. holds the Lee and William Abramowitz Professorial Chair of Biophysics.

## Author contributions

A.C. performed the experiments, analyzed the data, developed the theory, and wrote the simulation model. T.T. built the robotic platform. Together, A.C. and T.T. calibrated the platform and wrote the control scripts. N.S.G. helped with the theory and the simulation model. O.F. conceived the study and provided funds to conduct research. A.C., N.S.G., and O.F. wrote the manuscript. N.S.G. and O.F. supervised the project.

## Competing interests

The authors declare no competing interests.
