## [Transparent Peer Review file · Nature Communications]

Maximal response to a mechanical leader at critical group size in ant collectives

Corresponding Author: Dr Atanu Chatterjee

Version 0:

Reviewer comments:

Reviewer #1

(Remarks to the Author)

The authors have addressed all the comments and criticisms raised by the referees in their reports. In the revised version they have i) toned down the claims about criticality; ii) added more discussion about existing literature on response behavior; iii) clarified several points connected to the consistency between theory and experiments; iv) added new figures and text on the description of the phase transition, which help the reader to better understand the system. The manuscript reports novel and nice results on response behavior in ants, and the presentation is now balanced and detailed. The paper is therefore suitable for publication in Nature Communications.

(Remarks on code availability)

Reviewer #2

(Remarks to the Author)

The authors made substantial revisions based on my first comments, which I appreciate and for most part I am satisfied with.

However, I see an issue with the smoothed switching probabilities in Fig. 4, or at least there discrepancies between the smoothed data and the raw scatter data that to me do not make sense. My guess it is related to how the smoothing treats missing data in the bin in particular for low Δt .

For example for the small system there are no instances of resisting (failed switching), yet the heatmap shows a low switching probability (blue/greenish region) for low Δt . Similar phenomenon can be observed for low Δt for all system sizes. It becomes clearly apparent in the very large system, where for forces exceeding 0.4 there are no points where resisting is recorded. Thus I would expect a solid red color along the y-axis, yet the heatmap exhibits in this region a clear gradient in the switching probability along the y-axis. Thus, I do not trust the coarse-graining results as they are currently shown. This needs to be clarified.

(Remarks on code availability)

Reviewer #3

(Remarks to the Author)

In this paper, the authors study the dynamics of ant collective transport near the transition point between ordered and disordered motion, specifically measuring susceptibility to external forces. The work builds upon earlier papers from the same research group that had shown the existence of a hopf bifurcation in their real system, and here they demonstrate that

susceptibility to an applied external force is maximum near the “critical group size”. As we had noted, the study involving robots and ants is novel and perhaps the first to demonstrate maximum susceptibility at the critical point in a collective behaviour system.

1) Our main concern which we raised in our previous review was how relevant is to pitch the paper as adding broadly to criticality in biological systems.

(a) Since the number of agents are really small here, whether true phase transition properties (such as diverging correlation time/lengths, power-law scaling relations) are possible even in idealised circumstances is not clear. This issue was also raised by other reviewers. Authors sort of brush this aside saying that even if not all properties can be seen in their system – its indicate of phase transition. While I do agree that not all properties even needs to be expected to be measured in one research paper, I am not at all convinced that even if in principle these systems are to be seen as phase transitions.

(b) Secondly, but related to the same point, is that authors do previously show this to be a hopf-bifurcation system, as they do say so in one of the responses to our previous review.

(c) More concretely, the (normal forms at the mean-field level) dynamics of hopf bifurcation model and either continuous/discontinuous transitions are fundamentally different. Despite this, they can show similar properties such as vanishing eigen value or return rate near the bifurcation/critical points. In fact, as I had remarked in my previous review, literature on bifurcation-like transitions in ecology also shows evidence of many properties such as critical slowing down, even in systems without true phase transitions.

Therefore, we suggest authors to rather rephrase this manuscript Intro/Discussion as a generic disorder to order transition, without being too wedded to the idea of phase transitions in the physics sense; avoid overstating the results related to criticality, which this manuscript does not adequately addresses.

Note that we do not have any problems with the novelty of the set up and results, but it is the phrasing and interpretation of the results as evidence for criticality that I object to, and authors have not given convincing response on this point.

2) One could also imagine there being a stochastic mean-field approximation of the model, where the model always contains a deterministic limit cycle, but at smaller group sizes, the stochasticity is too large and 'blurs' the effects of the oscillations. The 'transition' in this case is a noise-induced transition with not even a true bifurcation.

To be specific: consider a non-linear oscillator with a noise term with a $1/\sqrt{N}$ dependence. For low N , the noise will obscure the oscillatory dynamics, and the histograms of x or v will show a single peak. As N increases, the histogram transitions to being bimodal. There is no bifurcation or deterministic transition here at all. How do we know this model does not explain the data shown by authors?

3) We would like to point out that although the authors claim that the data and code are available in a GitHub repo (<https://github.com/ant-lab-hub/robot-leader>), at the time of review, the linked repository seems to be empty (except for a readme file).

(Remarks on code availability)

Codes/Data were not available at the time of review. We were very keen to look at the code and check reproducibility, but unfortunately it was not available despite authors providing a link to github repository.

Reviewer #4

(Remarks to the Author)

(Remarks on code availability)

Version 1:

Reviewer comments:

Reviewer #3

(Remarks to the Author)

Authors have considered our comments seriously and updated the manuscript accordingly. We are satisfied with the responses and the changes made.

Code review

We thank the authors for clarifying the code repo, we are now able to access the code and datasets.

The following are our comments on the code and dataset:

Stochastic Model: There is a minor bug on the stochastic-model branch; line 3 of `ant_pendulum_run_open.py` should be `from ant_pendulum_open import *`. Once this was corrected, we were able to run the code without issues. Overall, the model code is clear to understand and is well-documented with comments.

Mean field Model: The code runs without issues. The code is reasonably clear to understand, but more documentation (like in the stochastic-model) could be helpful. Specifically, a clearer documentation of what each of the variables that are finally

being saved are, will be helpful.

Data: The datasets provided were a little difficult to understand, better documentation of the different variables (start_time, end_time, max_angleDiff, response) could be helpful.

In our view, there should also be clarification (and if possible, code examples) about which figures from the paper can be reproduced with the provided data/codes. As it stands, a reader will have to work hard to reproduce specific figures related to experimental data analysis and may not be sure if they got it right in the absence of detailed documentation. This is a point I would urge authors to consider provide clearer documentations, so that more readers can make use of data for newer analyses.

(Remarks on code availability)

We thank the authors for clarifying the code repo, we are now able to access the code and datasets.

The following are our comments on the code and dataset:

Stochastic Model: There is a minor bug on the stochastic-model branch; line 3 of ant_pendulum_run_open.py should be `from ant_pendulum_open import *`. Once this was corrected, we were able to run the code without issues. Overall, the model code is clear to understand and is well-documented with comments.

Mean field Model: The code runs without issues. The code is reasonably clear to understand, but more documentation (like in the stochastic-model) could be helpful. Specifically, a clearer documentation of what each of the variables that are finally being saved are, will be helpful.

Data: The datasets provided were a little difficult to understand, better documentation of the different variables (start_time, end_time, max_angleDiff, response) could be helpful.

In our view, there should also be clarification (and if possible, code examples) about which figures from the paper can be reproduced with the provided data/codes. As it stands, a reader will have to work hard to reproduce specific figures related to experimental data analysis and may not be sure if they got it right in the absence of detailed documentation. This is a point I would urge authors to consider provide clearer documentations, so that more readers can make use of data for newer analyses.

Reviewer #4

(Remarks to the Author)

(Remarks on code availability)

Response to reviewers

Reviewer 1

The authors have addressed all the comments and criticisms raised by the referees in their reports. In the revised version they have i) toned down the claims about criticality; ii) added more discussion about existing literature on response behavior; iii) clarified several points connected to the consistency between theory and experiments; iv) added new figures and text on the description of the phase transition, which help the reader to better understand the system. The manuscript reports novel and nice results on response behavior in ants, and the presentation is now balanced and detailed. The paper is therefore suitable for publication in Nature Communications.

We sincerely thank the reviewer for their positive assessment of our revised manuscript and their recommendation for publication.

Reviewer 2

The authors made substantial revisions based on my first comments, which I appreciate and for most part I am satisfied with.

We appreciate the reviewer's acknowledgment of our substantial revisions and are pleased they find our response satisfactory.

However, I see an issue with the smoothed switching probabilities in Fig. 4, or at least there are discrepancies between the smoothed data and the raw scatter data that to me do not make sense. My guess it is related to how the smoothing treats missing data in the bin in particular for low Δt .

For example for the small system there are no instances of resisting (failed switching), yet the heatmap shows a low switching probability (blue/greenish region) for low Δt . Similar phenomenon can be observed for low Δt for all system sizes. It becomes clearly apparent in the very large system, where for forces exceeding 0.4 there are no points where resisting is recorded. Thus I would expect a solid red color along the y-axis, yet the heatmap exhibits in this region a clear gradient in the switching probability along the y-axis. Thus, I do not trust the coarse-graining results as they are currently shown. This needs to be clarified.

We acknowledge the reviewer's concern regarding the apparent 'discrepancy' between the binary scatter plots in the SI and the smoothed heatmap in Fig. 4 (Main Text). As noted in the "Materials and Methods: Data Analysis" section, the scatter plots show raw, per-trial

switch or no-switch events, whereas Fig. 4 displays mean switching probabilities after binning and smoothing. A key point is that if a force F_{ext} is applied at $t = 0$ and we measure a switch at $t = T$, then by definition, the system did *not* switch at any $t < T$. Therefore, each observed switch at time t_{switch} contributes a “no-switch” record for all times $t < t_{\text{switch}}$. Even in the small-system example where every trial *eventually* switches, most of those switches occur at $t \gtrsim 1$ s, which gives rise to a low switching probability at shorter times, as pointed out by the reviewer.

This is clearly explained in lines 330-338 in the main text, as reproduced here below: “For a given magnitude of external force, F_{ext} , a group may either resist the force or switch its alignment in response. To determine the switching probability per time bin, the entire time range – from 0 to the maximum duration of force application observed across all trials for that force – is first divided into evenly spaced bins. The duration of force application in each trial is then mapped onto these bins. In a particular trial, for a given external force magnitude, if the cargo does not switch throughout the entire duration ($\Delta t = t_{\text{end}} - t_{\text{begin}}$), then all corresponding bins are recorded as ‘no switch’ bins. However if the cargo does switch its direction of motion at time t_{switch} , we record all bins prior to that moment as no switch”, and the bin in which the switch occurs ($\Delta t = t_{\text{switch}} - t_{\text{begin}}$) is marked as a switch bin”. The switching probability per time bin for each group size is then represented as heatmaps (shown in Fig.4), generated by binning the data using a k -nearest neighbor approach and smoothing with a simple moving average.”

Reviewers 3 and 4

In this paper, the authors study the dynamics of ant collective transport near the transition point between ordered and disordered motion, specifically measuring susceptibility to external forces. The work builds upon earlier papers from the same research group that had shown the existence of a hopf bifurcation in their real system, and here they demonstrate that susceptibility to an applied external force is maximum near the “critical group size”. As we had noted, the study involving robots and ants is novel and perhaps the first to demonstrate maximum susceptibility at the critical point in a collective behaviour system. 1) Our main concern which we raised in our previous review was how relevant is to pitch the paper as adding broadly to criticality in biological systems. (a) Since the number of agents are really small here, whether true phase transition properties (such as diverging correlation time/lengths, power-law scaling relations) are possible even in idealised circumstances is not clear. This issue was also raised by other reviewers. Authors sort of brush this aside saying that even if not all properties can be seen in their system – its indicate of phase transition. While I do agree that not all properties even needs to be expected to be measured in one research paper, I am not at all convinced that even in principle these systems are to be seen as phase transitions.

We appreciate the reviewers’ caution regarding the use of “phase transition” in our relatively small ant collectives. Indeed, classical phase transitions, in the strict sense, generally require infinite system sizes. However, our spin-based model (Fig. 1a and 1b) predicts that even finite ant groups can display a finite-size transition between two collective states (order and disorder). We showed that the system’s order-disorder transition can be traversed by adjusting either the interaction strength between spins (the “individuality parameter” in Fig. 1c) or

Figure 1: (a,b) Simulated cargo response to a single informed ant attaching. (a) Response as a function of the individuality parameter F_{ind} . Insets show velocity distributions for small (orange) and large (blue) F_{ind} and the corresponding best-fit value (pink). The upper-left inset depicts representative trajectories (all starting at the yellow dot; color-coded as in the inset) that account for continual arrivals of informed ants (scale bar = 10 cm). (b) Response as a function of cargo radius, where the pink dot indicates the experimental load radius. (c,d) Exact solution of the Ising spin model. (c) Normalized system response versus F_{ind} ; the blue curve is the short-term response to a newly attached ant, and the red curve is the mean-field response, which diverges at the critical point. (d) Normalized short-term response versus the mean number of ants attached (a proxy for load size). Dotted lines mark the critical transition points (*Nat. Comm.*, 6(1), 7729, (2015)).

the number of spins (the system size, Fig. 1d). Both mean-field calculations and numerical simulations show a pronounced peak in responsiveness at an intermediate group size, consistent with our experimental observations of short-time responses. While we acknowledge the limitations posed by the finiteness of our system, analyzing these phenomena through the lens of criticality offers valuable insights into the functionality of collective behavior in organisms. We have revised our text throughout to clarify that we do not claim a strict phase transition in the thermodynamic sense (see below), and we have added a paragraph in the Discussion to highlight this distinction further (see lines 273-288):

“When longhorn crazy ants engage in cooperative transport, each ant either pulls on the load or lifts it. These binary choices can be described using a spin-like (Ising-type) model. The fact that all ants are connected and influence each other through a shared cargo, effectively translates into all-to-all interactions. The resulting model, in its thermodynamic limit, predicts an order-disorder phase transition as a function of the coupling parameter, F_{ind} . We have further shown that the experimentally modifiable group size can serve as a control pa-

parameter instead of the coupling strength [10]. We theoretically demonstrated that the ant collective indeed undergoes an order-to-disorder transition as a function of group size [10,11]. In agreement, the motion of the load carried by ants switches from ballistic motion to a biased random walk at the predicted group size [31]. The thermodynamic limit of the model further tells us that at this critical transition point, the susceptibility to small external fields diverges. Experimentally, we can only measure short-time responses, specifically, how much the ant group moves in response to an applied force. Our analytical mean-field solution and numerical simulations of the Ising model on finite sizes predict that, in this case, a peak in responsiveness will be evident at the phase transition (critical group size). As reported in this work, this peak is indeed observable in a group of carrying ants. Therefore, although criticality and divergence of susceptibility are usually defined in the thermodynamic limit of large systems, our empirical results, together with our microscopically grounded model, establish a clear link between the phase transition that the model predicts in the thermodynamic limit and the finite-size order-disorder transition observed in real ant groups.”

We have also added the rationale behind measuring short-time response (see lines 328-333):

“Each nudge was kept short relative to the natural oscillation period of the cargo. Applying force over a longer interval would significantly change the cargo angle during the push, thereby altering the system conditions mid-measurement and obscuring our ability to observe the short-time response under a fixed set of parameters. The short-duration forcing thus allows us to measure how the ant group responds under near-instantaneous, controlled perturbations, rather than confounding that response with the changing rod position.”

(b) Secondly, but related to the same point, is that authors do previously show this to be a hopf-bifurcation system, as they do say so in one of the responses to our previous review.

We appreciate the reviewers’ thoughtful comment regarding the link between the phase transition in our cooperative transport model and the Hopf bifurcation in the hinged load’s oscillatory dynamics. Our main objective is to understand how individual-ant dynamics give rise to collective states at the group scale. In particular, our spin-based cooperative transport model predicts an order–disorder transition as a function of group size, manifested by a pronounced peak in the ants’ response to an external force at a critical size. In our previous study, *PNAS* 113 (51) 2016, we showed that one could derive a simplified, one-dimensional mean-field description of the hinged load’s motion from the microscopic model of individual ants. In this reduced set of equations, the order-disorder phase transition takes the form of a Hopf bifurcation in the hinged load’s dynamics, even though the ants themselves no longer appear explicitly in this simplified description. Thus, the Hopf bifurcation is not an alternative explanation but rather an emergent manifestation of the same underlying collective model.

(c) More concretely, the (normal forms at the mean-field level) dynamics of hopf bifurcation model and either continuous/discontinuous transitions are fundamentally different. Despite this, they can show similar properties such as vanishing eigen value or return rate near the bifurcation/critical points. In fact, as I had remarked in my previous review, literature on bifurcation-like transitions in ecology also shows evidence of many properties such as critical slowing down, even in systems without true phase transitions. Therefore, we suggest authors to rather rephrase this manuscript Intro/Discussion as a generic disorder to order transition, without being too wedded to the idea of phase transitions in the physics sense; avoid overstating the results related to criticality, which this manuscript does not adequately addresses.

Figure 2: Mean velocity of the one-dimensional motion calculated using the simplified coupled model plotted against F_{ind}/N . The solid curves are the analytical solutions of the mean-field equation while the circles represent averages obtained from simulations with different group sizes, N (*Nat. Phys.*, 14, 683–693, 2018).

Note that we do not have any problems with the novelty of the set up and results, but it is the phrasing and interpretation of the results as evidence for criticality that I object to, and authors have not given convincing response on this point.

We thank the reviewer for recognizing the novelty of our setup and results. In light of the concerns raised regarding the terms “criticality” and “phase transitions,” we agree with the reviewers’ suggestion and have revised our terminology to avoid overstating our conclusions regarding strict phase transitions. Below is a list of changes we have implemented in the revised manuscript:

- Abstract – divergence → peak
- line 32 near the critical point → close to the transition regime
- line 34 critical → critical-like
- line 38 critical → transition between collective states
- line 49 “...order-disorder...”
- line 55 “...which represent the ordered and disordered phases of the ants, respectively,...”
- line 59 critical → transition
- line 69 near and far from criticality → across various collective states
- line 259 critical → transition
- line 305 near a phase-transition regime → close to a transition regime
- line 308 divergence → peak
- line 308 critical → intermediate

2) One could also imagine there being a stochastic mean-field approximation of the model, where the model always contains a deterministic limit cycle, but at smaller group sizes, the stochasticity is too large and ‘blurs’ the effects of the oscillations. The ‘transition’ in this case is a noise-induced transition with not even a true bifurcation. To be specific: consider a non-linear oscillator with a noise term with a $1/\sqrt{N}$ dependence. For low N , the noise will obscure the oscillatory dynamics, and the histograms of x or v will show a single peak. As N increases, the histogram transitions to being bimodal. There is no bifurcation or deterministic transition here at all. How do we know this model does not explain the data shown by authors?

We appreciate the reviewer for raising this concern. While a noise-driven oscillator could indeed capture certain macroscopic oscillatory dynamics, our approach differs fundamentally by operating at the scale of individual ants. Each ant can only pull or lift, naturally mapping its discrete behavior to a spin-like (Ising-type) model. This microscopically grounded framework requires minimal fitting and has been extensively validated under a range of experimental conditions – spanning free cargo transport (*Nat. Comm.* 6(1):7729, 2015) and confined transport (*PLoS Comput. Biol.* 14(5):e1006068, 2018; *PNAS* 122(1):e2414274121, 2025).

Both our model and experiments show that varying the group size N drives a transition between ordered and disordered collective states (Fig. 2). As N grows very large (approaching the theoretical infinite limit), this transition becomes sharp and exhibits hallmarks of critical behavior. At lower N , the transition is less pronounced, reflecting finite-size effects, yet upon proper scaling, it still aligns with the large N results, as it is simply a finite-size realization of the same critical phenomenon.

In conclusion, by grounding our model in directly observed microscopic rules and verifying it under multiple conditions, we show how local spin-like decisions give rise to collective emergent oscillations. This track record of predictive success distinguishes our approach from purely phenomenological noise-driven oscillator models that do not incorporate the discrete pull-or-lift constraint. Our findings thus offer robust evidence for a genuine, if finite-size, order-disorder transition in ant transport behavior.

3) We would like to point out that although the authors claim that the data and code are available in a GitHub repo (<https://github.com/ant-lab-hub/robot-leader>), at the time of review, the linked repository seems to be empty (except for a readme file).

The simulation codes – mean-field model, agent-based Gillespie model, and raw data files are available in the GitHub repository as separate branches. To access these materials, please navigate to the GitHub repository and locate the “Branch” drop-down menu at the top-left of the file listing (which may default to “main” with the .readme file). Clicking the arrow will reveal branches labeled “mean-field-model,” “stochastic-model,” and “Data,” corresponding to the mean-field model, agent-based Gillespie model, and raw data files, respectively. We hope these instructions clarify how to locate the code and data in the repository, and we apologize once again for any confusion.

Response to reviewers

Reviewers 3 and 4

Authors have considered our comments seriously and updated the manuscript accordingly. We are satisfied with the responses and the changes made.

Code review

We thank the authors for clarifying the code repo, we are now able to access the code and datasets. The following are our comments on the code and dataset:

Stochastic Model

There is a minor bug on the stochastic-model branch; line 3 of `ant_pendulum_run_open.py` should be `from ant_pendulum_open import *`. Once this was corrected, we were able to run the code without issues. Overall, the model code is clear to understand and is well-documented with comments.

Mean field Model

The code runs without issues. The code is reasonably clear to understand, but more documentation (like in the stochastic-model) could be helpful. Specifically, a clearer documentation of what each of the variables that are finally being saved are, will be helpful.

Data

The datasets provided were a little difficult to understand, better documentation of the different variables (`start_time`, `end_time`, `max_angleDiff`, `response`) could be helpful.

In our view, there should also be clarification (and if possible, code examples) about which figures from the paper can be reproduced with the provided data/codes. As it stands, a reader will have to work hard to reproduce specific figures related to experimental data analysis and may not be sure if they got it right in the absence of detailed documentation. This is a point I would urge authors to consider provide clearer documentations, so that more readers can make use of data for newer analyses.

We appreciate the reviewers' positive feedback on our revisions. In response to their suggestions, we have fixed the bug in the stochastic-model code, added detailed documentation to the mean-field model code, and expanded the documentation for all data files.